# The relationship between interviewer-respondent familiarity and family planning outcomes in the Democratic Republic of Congo: a repeat cross-sectional analysis

Philip Anglewicz,[1] Pierre Akilimali,[2] Linnea Perry Eitmann,[3] Julie Hernandez,[3] Patrick Kayembe[4]

For numbered affiliations see end of article.

**Correspondence to**
Dr Philip Anglewicz;
panglewi@tulane.edu

## ABSTRACT

**Objectives** The typical approach of survey data collection is to use interviewers who are not from the study site and do not know the participants, yet the implications of this approach on data quality have seldom been investigated. We examine the relationship between interviewer–respondent familiarity and selected family planning outcomes, and whether this relationship changes over time between 2015 and 2016.

**Setting** We use data from the Performance Monitoring and Accountability 2020 Project in Kongo Central Province, Democratic Republic of Congo.

**Participants** Participants include representative samples of women of reproductive ages (15 to 49), 1565 interviewed in 2015 and 1668 in 2016. The study used a two-stage cluster design: first randomly selecting enumeration areas (EAs), then randomly selecting households within each EA.

**Design** We first identify individual characteristics associated with familiarity between RE and respondent. Next, we examine the relationship between RE–respondent acquaintance and family planning outcomes. Finally, we use two waves of data to examine whether this relationship changes over time between 2015 and 2016.

**Results** In multivariate analysis, interviewer–respondent acquaintance is significantly associated with last birth unintended (OR 1.91, 95% CI 1.17 to 3.13) and reported infertility in 2015 (OR 2.26, 95% CI 1.03 to 4.95); and any contraceptive use (OR 1.51, 95% CI 1.01 to 2.28), traditional contraceptive use (OR 1.79, 95% CI 1.10 to 2.89), reported infidelity (OR 1.89, 95% CI 1.02 to 3.49) and age at first sex (coefficient −0.48, 95% CI −0.96 to −0.01) in 2016. The impact of acquaintance on survey responses changed over time for any contraceptive use (OR 2.09, 95% CI 1.33 to 3.30).

**Conclusions** The standard in many large-scale surveys is to use interviewers from outside the community. Our results show that interviewer–respondent acquaintance is associated with a range of family planning outcomes; therefore, we recommend that the approach to hiring interviewers be examined and reconsidered in survey data collection efforts.

## Strengths and limitations of this study

► To examine a rarely studied issue in research methods: the relationship between interviewer–respondent familiarity and survey responses.

► Investigating the change in the relationship between interviewer–respondent familiarity and survey responses over time.

► Lack of randomised controlled trial study design, and lack of validation of survey responses.

## BACKGROUND

The interviewer is a potential source of error in survey research.[1] Ideally, there should be no systematic variation in survey responses by interviewer, particularly if interviewers are randomly assigned to respondents.[2] Yet interviewers can influence survey responses in many ways. The influence can be subtle, in which interviewers' behaviours, body language and appearance during the interview impacts survey responses.[3 4] Or respondents may be affected by basic characteristics of interviewers: research on 'interviewer effects' often examines the impact of interviewers' age, gender, race, physical appearance, work experience and the comparison of these characteristics between interviewer and respondent.[2 5–10]

Studies often show that interviewers systematically influence survey responses.[2] This potential source of bias is often ignored entirely. Other times, interviewer characteristics are seen as part of the error term in multivariate regression and assumed to not impact estimates, despite the fact that these characteristics may in fact be associated with outcomes of interest, and descriptive statistics (like contraceptive use) are often of primary interest.

While the vast majority of research on this topic has focused on demographic characteristics of interviewers, there are other characteristics of interviewers that have been studied less frequently, such as the degree of familiarity between survey interviewer and respondent. The limited research on interviewer–respondent familiarity has shown that it affects survey responses.[11–13]

But the nature of the influence is not clear: does familiarity between interviewer and respondent lead to more accurate responses or worse data quality? Theory suggests that either is possible. On one hand, greater familiarity may make respondents less forthcoming, due to fear of the interviewer disclosing sensitive information to mutual acquaintances, or fear of judgement from a respected peer.[14 15] For these reasons, data collection techniques that reduce the role of the interviewer in survey administration (eg, ballot box, ACASI, CASI) have been used for surveys involving sensitive behaviours.[16–18] However, many societies view outsiders with suspicion and may therefore less likely provide accurate responses to interviewers who are 'outsiders'.[12 13] In addition, understanding of local customs may improve communication between interviewer and respondent.[13] The relatively limited testing on the relationship between interviewer–respondent familiarity and survey responses suggests that data quality may improve with a greater degree of familiarity.[11–13] But this relationship has seldom been examined, and research suggests that the impact of interviewer–respondent familiarity likely varies by context and over time.[12 13]

We use data from a study where the role of interviewers is particularly important for data collection. Unlike the typical survey data collection approach of using 'outsider' or 'stranger' interviewers who have little or no connection with the setting for data collection, like Demographic and Health Surveys (DHS), the Performance Monitoring and Accountability 2020 Project (PMA2020) recruits interviewers from each enumeration area (EA) in their samples.[19] Since PMA2020 interviewers live in the EA, they are referred to as 'resident enumerators' (REs).

Using data from PMA2020 in the Democratic Republic of Congo (DRC), we examine the relationship between RE–respondent acquaintance and survey responses. We first identify individual characteristics associated with greater familiarity between RE and respondent. Next, we conduct bivariate and multivariate tests of association between RE–respondent acquaintance and survey outcomes of particular interest to the PMA2020 study, including contraceptive use, whether last birth was unintended, reporting child mortality, reporting infertility, providing a response to age at sexual debut and the age at sexual debut. Finally, we use two waves of PMA2020 data to examine if the relationship between RE–respondent acquaintance changes over time between the first two waves of PMA2020 in 2015 and 2016.

## METHODS
### Setting
DRC is Africa's third most populous and one of the region's fastest growing countries.[20] DRC has one of the highest fertility rates in the world: the most recent DHS from 2013 to 2014 estimated a country-level TFR of 6.6, a slight increase since the 6.3 TFR estimated from the 2007 DHS.[21 22] At the same time, contraceptive use is low in DRC: the modern contraceptive prevalence rate (mCPR) among women aged 15–49 who are married or in union is 7.8% for the country as a whole, 19.0% in Kinshasa and 17.2% in Kongo Central.[22]

### Data performance, monitoring and accountability 2020
PMA2020 was established in part to measure uptake of contraceptive use in many of the world's most populous countries (http://www.pma2020.org/). To achieve this aim, PMA2020 collects representative data in 11 countries on an annual basis for a range of fertility and family planning-related measures.

One of the important features of the PMA2020 approach to data collection is the use of 'Resident Enumerators'. PMA2020 recruits women to collect data from their own enumeration area. So in contrast to most data collection efforts, where most interviewers will not know respondents (or vice versa), it is not unlikely that some respondents and REs are acquainted with one another.

PMA2020 has several requirements for recruitment of REs. REs should be over age 21 and hold a high school diploma or higher level of educational attainment. REs should not have an affiliation with the local health system. Since PMA2020 collects data via mobile phone, REs are recommended to have some degree of familiarity with use of mobile phones. Due to these requirements, REs are typically recruited from schools or other governmental entities in PMA2020 EAs.

To date, PMA2020 has collected data from two provinces in DRC, five rounds of data in Kinshasa (2013–2016), and two rounds in Kongo Central (2015–2016), a province to the west of Kinshasa. The sampling framework uses a two-stage cluster sampling approach, in which the study first randomly selects census enumeration areas within each province, then conducts a listing of all households in these EAs and randomly selects 33 households within each EA. The enumeration areas remain the same over time, but new households are selected in each round. Participating households were selected by the central survey management team, not the RE, which ensures that REs did not systematically select households with friends or acquaintances. PMA2020 first administers a household survey to the head of household, and then all resident women of reproductive age (15–49 years) within the household are selected for interview. The PMA2020 female survey includes basic demographic information and extensive information on fertility history and preferences, and contraceptive use.

## Measures

Our measure of primary interest in this analysis is acquaintance between the RE and respondent, which is measured by the question 'How well acquainted are you with the respondent?' (response options 'very well acquainted', 'well acquainted', 'not well acquainted' and 'not acquainted'). The RE completes this question at the beginning of the female survey. Acquaintance with the respondent was collected in surveys for all PMA2020 countries, and was defined in the PMA2020 RE training manual: 'very well acquainted' was defined as the RE knowing the respondent's first name and would greet her if they met at the market, church or mosque; 'well acquainted' was that the RE knows the respondent by sight and may know a family member as well; 'not well acquainted' is that the RE has seen the woman at community or church functions but does not know her name or does not recognise her but knows someone else in the household; 'not acquainted' as the RE has never seen the respondent or anyone in her family before. A second measure of interest is whether the respondent was previously interviewed, collected during the second wave of PMA2020 in 2016.

Due to the densely populated urban setting, there was not sufficient variation in the measure of familiarity between REs and respondents in Kinshasa (98.9% were 'not acquainted' in 2015). Thus, in this research, we use only data from the two rounds of data collection in Kongo Central. Data collection for round 1 (R1) in Kongo Central was conducted from November 2015 to January 2016; the household response rate was 96.3% and yielded 1625 households in which 1565 women were interviewed (95.8% response rate). Round 2 (R2) was conducted in August and September of 2016, and 1668 women (96.9% response rate) were interviewed from among 1575 households (response rate 96.0%).

Our outcome measures were selected for three reasons. First, they are of primary interest to the PMA2020 study. Since PMA2020 focuses on changes in family planning indicators over time, we include measures of contraceptive use (overall contraceptive use, use of modern methods, use of traditional methods), and whether the last birth was unintended (wanted later or not at all). Second, we identify several measures in the PMA2020 survey that are particularly sensitive and therefore may be more influenced by RE behaviour, including providing a response to sexual debut (as opposed to refusing to respond or stating 'don't know'), age of sexual debut, experiencing the death of a child and reporting 'infertility' as the reason for not wanting more children. Third, most of these measures are of interest to a range of research topics, such as family planning, reproductive health, sexual behaviour, HIV/AIDS and fertility.

For our multivariate analysis, we also consider measures that are likely to impact familiarity between RE and respondent, including age, level of education, urban/semiurban/rural residence, marital status and number of lifetime births. We include a measure of household wealth, created using a wealth index based on ownership of 25 household durable assets, and created using principal component analysis,[23] then converted into quintiles.

All participants were randomly selected to participate in the study. We presented the goals of the study to all eligible participants. Before the survey instrument was administered, informed consent was obtained from all participants. PMA2020 has received approval to collect data from Institutional Review Boards at Johns Hopkins University, Tulane University, and the University of Kinshasa.

## Patient and public involvement

Neither patients nor public were involved in study design or conduct of the study, and there are no set plans to disseminate the results to participants.

## Analysis

We conduct our analysis in four steps. First, we identify characteristics associated with greater familiarity between RE and respondent. To do so, we use ordered logistic regressions where the dependent variable is the four-category variable for level of acquaintance, and independent variables include age, a quadratic term for age, level of education, marital status, household wealth quintile and urban/rural residence. To account for the study design, we cluster standard errors by enumeration area. We run these regressions separately for each of the two waves of PMA2020 data in Kongo Central, 2015 and 2016.

Next, we examine whether acquaintance between REs and respondents potentially affects survey responses. We begin with bivariate associations between the four-category measure of acquaintance and the outcome measures above. We use $\chi^2$ tests and t-tests to examine whether the values of each outcome are significantly different for those who are 'not well acquainted,' 'well acquainted' and 'very well acquainted' compared with those who are 'not acquainted'. After the bivariate tests, we run multivariate regressions where the dependent variables are the eight outcomes listed above. Independent variables of primary interest are the four-category measure of acquaintance and previously interviewed by PMA2020 (included only for 2016). Other independent variables include age, level of education, marital status, number of lifetime births, urban/rural residence and wealth quintile. As previously, we cluster SE by enumeration area. We run ordinary least squared regression for age at sexual debut and logistic regression for all other outcome measures (separately by PMA2020 survey wave).

Finally, we examine whether the impact of RE–respondent acquaintance on survey outcomes changes over time. To do so, we pool data for both waves (2015 and 2016), and generate a binary indicator for the second wave. We then create interaction terms between the second survey wave and all acquaintance measures to examine if the relationship between acquaintance and the dependent variables differs by year. We again cluster SE by EA.

**Table 1** Background characteristics, Performance Monitoring and Accountability 2020 Project women in Kongo Central, rounds 1 (2015) and 2 (2016)

| | 2015 | 2016 |
|---|---|---|
| RE–respondent acquaintance | | |
| Very well acquainted | 13.9% | 12.7% |
| Well acquainted | 25.3% | 21.5% |
| Not well acquainted | 19.7% | 19.8% |
| Not acquainted | 41.1% | 46.0% |
| Interviewed in previous wave | – | 10.8% |
| Age (mean, SD) | 22s.8, 0.22 | 23.1, 0.22 |
| Number of births (mean, SD) | 2.4, 0.06 | 2.4, 0.06 |
| Education | | |
| No school | 10.8% | 11.2% |
| Primary | 31.8% | 32.2% |
| Secondary | 54.2% | 54.5% |
| Tertiary or higher | 3.2% | 2.1% |
| Marital status | | |
| Currently married | 36.1% | 34.7% |
| Coresiding but not married | 29.5% | 23.7% |
| Divorced | 7.0% | 7.2% |
| Widowed | 1.2% | 1.4% |
| Never married | 26.2% | 32.9% |
| Wealth quintile | | |
| Lowest quintile | 16.6% | 14.0% |
| Lower quintile | 18.1% | 14.0% |
| Middle quintile | 21.2% | 18.2% |
| Higher quintile | 20.8% | 23.2% |
| Highest quintile | 23.3% | 30.6% |
| Urban/rural residence | | |
| Rural | 55.6% | 56.2% |
| Semiurban | 19.3% | 18.3% |
| Urban | 25.1% | 25.5% |
| Outcome measures | | |
| Using any contraceptive method | 29.8% | 34.4% |
| Using a modern contraceptive method | 19.9% | 19.2% |
| Using a traditional contraceptive method | 9.8% | 15.2% |
| Last birth was unintended | 17.0% | 15.7% |
| Experienced the death of a child | 25.2% | 25.8% |
| Provided a response to sexual debut | 83.5% | 80.3% |
| Reported infertility | 9.3% | 10.3% |
| Age at first sex (mean, SD) | 16.3, 0.08 | 15.9, 0.07 |
| N | 1565 | 1668 |

**Table 2** Ordered logistic regression results for characteristics associated with acquaintance between RE and respondent, PMA2020 Kongo Central 2015, 2016

| | R1 (2015) | R2 (2016) |
|---|---|---|
| | OR (95% CI) | OR (95% CI) |
| Age | 0.92s (0.83 to 1.02) | 1.02* (1.01 to 1.04) |
| $Age^2$ | 1.00 (0.99 to 1.00) | – |
| Number of births | 1.06 (0.95 to 1.18) | 1.03 (0.94 to 1.14) |
| Interviewed in previous wave | – | 2.82** (1.64 to 4.86) |
| Education | | |
| No school (reference) | – | – |
| Primary | 1.20 (0.60 to 2.41) | 0.97 (0.61 to 1.53) |
| Secondary | 1.41 (0.68 to 2.94) | 1.20 (0.72 to 2.00) |
| Tertiary | 1.02 (0.40 to 2.58) | 1.24 (0.59 to 2.58) |
| Marital status | | |
| Never married (reference) | – | – |
| Currently married | 1.83 (1.12 to 3.00) | 0.77 (0.51 to 1.17) |
| Coresiding but not married | 0.97 (0.63 to 1.49) | 0.85 (0.51 to 1.42) |
| Divorced | 0.72 (0.39 to 1.33) | 0.63 (0.35 to 1.13) |
| Widowed | 1.21 (0.40 to 3.65) | 0.91 (0.30 to 2.72) |
| Wealth quintile | | |
| Lowest quintile (reference) | – | – |
| Lower quintile | 1.19 (0.63 to 2.25) | 0.62 (0.31 to 1.26) |
| Middle quintile | 1.44 (0.69 to 3.00) | 0.77 (0.37 to 1.63) |
| Higher quintile | 0.86 (0.35 to 2.11) | 0.66 (0.28 to 1.56) |
| Highest quintile | 0.91 (0.30 to 2.79) | 0.68 (0.25 to 1.83) |
| Residence | | |
| Urban (reference) | – | – |
| Rural | 4.38 (1.51 to 12.68) | 4.48 (1.46 to 13.76) |
| Semiurban | 2.72 (0.82 to 9.09) | 2.68 (0.85 to 8.48) |
| N | 1565 | 1668 |

*p<0.05, **p<0.01.
PMA2020, Performance Monitoring and Accountability 2020 Project; RE, resident enumerator.

## RESULTS

Background characteristics for women from both rounds of PMA2020 are shown in table 1. Table 2 shows results for characteristics associated with acquaintance between RE and respondent. In both waves, rural residence is associated with significantly greater odds of acquaintance than urban residence, with an OR of 4.38 in 2015 (95% CI 1.51 to 12.68) and 4.48 (95% CI 1.46 to 13.76) in 2016. In the first wave, marital status is significantly associated with acquaintance; with acquaintance between RE and respondent more likely among women who are currently married than the never married. In 2016, there is a positive association between age and acquaintance and, not surprisingly, those interviewed in 2015 are significantly more likely to be acquainted than those who were not interviewed.

**Table 3** Percentage of respondents with family planning outcomes by each category of RE-respondent acquaintance, PMA2020 Kongo Central 2015–2016

| | Not acquainted | Not well acquainted | Well acquainted | Very well acquainted | Total |
|---|---|---|---|---|---|
| **Round 1 2015** | | | | | |
| Using any contraceptive method | 34.5% | 33.5% | 23.1%** | 21.5%** | 29.8% |
| Using a modern contraceptive method | 23.2% | 23.2% | 14.8%** | 14.2%** | 19.9% |
| Using a traditional contraceptive method | 11.3% | 10.2% | 8.3% | 7.3% | 9.8% |
| Last birth was unintended | 60.1% | 67.3% | 76.5%** | 67.3% | 66.6% |
| Experienced the death of a child | 23.3% | 20.0% | 30.3%* | 27.5% | 25.1% |
| Provided a response to sexual debut | 80.0% | 83.8% | 88.5%** | 85.8% | 83.5% |
| Reported infertility | 7.3% | 7.8% | 10.2% | 16.0%** | 9.3% |
| Age at sexual debut (years) | 16.7 | 16.2 | 16.2 | 15.7** | 16.3 |
| **Round 1 2016** | | | | | |
| Using any contraceptive method | 34.2% | 37.9% | 34.9% | 29.1% | 34.4% |
| Using a modern contraceptive method | 20.6% | 22.8% | 16.5% | 13.2%* | 19.2% |
| Using a traditional contraceptive method | 13.6% | 15.1% | 18.5% | 16.0% | 15.2% |
| Last birth was unintended | 58.5% | 65.6% | 64.7% | 63.3% | 62.0% |
| Experienced the death of a child | 24.3% | 32.9%* | 20.4% | 28.4% | 25.8% |
| Provided a response to sexual debut | 77.2% | 82.5% | 82.1% | 84.8%* | 80.3% |
| Reported infertility | 7.0% | 11.0% | 14.4%* | 13.8%* | 10.3% |
| Age at sexual debut (years) | 16.2 | 15.9 | 15.6** | 15.6* | 15.9 |

Bivariate $\chi^2$ test or t-test of difference with 'not acquainted' category is statistically significant at *p<0.01, **p<0.001.
PMA2020, Performance Monitoring and Accountability 2020 Project; RE, resident enumerator.

Next, we tabulate levels of each outcome measure for each category of acquaintance for both 2015 and 2016 (table 3). In 2015, we find that RE and respondents who are well acquainted and very well acquainted are significantly less likely to be using any contraceptive method (23.1% and 21.5% vs 34.5%, p<0.01) and a modern method (14.8% and 14.2% vs 23.2%, p<0.01) than REs and respondents who are not acquainted. We also find that REs and respondents who are well acquainted are significantly more likely to report that the last birth was unintended (75.6% vs 60.1%, p<0.01), to have experienced the death of a child (30.3% vs 23.3%, p<0.05), and were more likely to provide a response to sexual debut (88.5% vs 80.0%, p<0.01). REs and respondents who are very well acquainted were more likely to report infertility (16.0% vs 7.3%, p<0.01) and had a significantly younger age at sexual debut (15.7 years vs. 16.7 years, p<0.01) than REs and respondents who were not acquainted.

In 2016, REs and respondents who were well acquainted were more likely to report infertility (14.4% vs 7.0%, p<0.05), and reported a significantly younger age at sexual debut (15.6 years vs 16.2 years, p<0.01). REs and respondents who were very well acquainted were less likely to report using a modern method (13.2% vs 20.6%, p<0.01), more likely to provide a response to sexual debut (84.8% vs 77.2%, p<0.05), more likely to report infertility (13.8% vs 7.0%, p<0.05) and reported a younger age at sexual debut (15.6 years vs 16.2 years, p<0.05). Traditional method use was the only outcome that was not

significantly different for well or very acquainted REs–respondents compared with not acquainted in either year.

After controlling for characteristics associated both with RE–respondent acquaintance and contraceptive use, such as age and urban residence, our results differ from those shown in table 2. Multivariate results that control for characteristics associated with acquaintance and our outcomes of interest are shown in table 4. Here, we again find some significant associations between outcomes of interest and acquaintance, particularly for REs and respondents who are well acquainted. REs and respondents who are well acquainted are significantly more likely to say that the most recent birth was unintended (in 2015, OR 1.91 (95% CI 1.17 to 3.13)), and are more likely to report using traditional methods (in 2016, OR 1.79 (95% CI 1.10 to 2.89)), more likely to report infertility (in 2016, OR 1.89 (95% CI 1.02 to 3.49)), and a lower age at sexual debut (in 2016, OR −0.48 (95% CI −0.96 to −0.01)), compared with REs and respondents who are not acquainted. Also in 2015, REs and respondents who are very well acquainted are more likely to report infertility than those who are not acquainted (OR 2.26 (95% CI 1.03 to 4.95)). It is important to note that previous interview was not significantly associated with any outcomes in 2016.

Finally, results for differences by survey wave in the relationship between RE–respondent acquaintance and our outcomes of interest are shown in table 5. The only interactions between year and acquaintance that are

**Table 4** Multivariate regression results for the relationship between RE–respondent acquaintance and family planning outcomes, PMA2020 Kongo Central 2015, 2016

| | Using any contraceptive method | Using a modern contraceptive method | Using a traditional contraceptive method | Last birth was unintended | Experienced the death of a child | Provided a response to sexual debut | Reported infertility | Age at first sex (mean) |
|---|---|---|---|---|---|---|---|---|
| | Odds 95% CI | Odds 95% CI | Odds 95% CI | Odds 95% CI | Odds 95% CI | Odds 95% CI | Odds 95% CI | Coef. 95% CI |
| **RE–respondent acquaintance** | Round 1 2015 | | | | | | | |
| Not acquainted (reference) | – | – | – | – | – | – | – | – |
| Not well acquainted | 1.11 | 1.18 | 0.94 | 1.36 | 0.75 | 1.30 | 1.08 | –0.32 |
| | (0.71 to 1.73) | (0.76 to 1.84) | (0.47 to 1.87) | (0.79 to 2.34) | (0.36 to 1.58) | (0.73 to 2.33) | (0.68 to 1.72) | (–1.10 to 0.46) |
| Well acquainted | 0.63 | 0.68 | 0.68 | 1.91** | 1.12 | 1.80 | 1.44 | –0.09 |
| | (0.34 to 1.15) | (0.36 to 1.28) | (0.31 to 1.51) | (1.17 to 3.13) | (0.53 to 2.37) | (0.94 to 3.45) | (0.80 to 2.60) | (–0.71 to 0.52) |
| Very well acquainted | 0.60 | 0.70 | 0.59 | 1.31 | 0.94 | 0.67 | 2.26* | –0.46 |
| | (0.29 to 1.24) | (0.34 to 1.44) | (0.24 to 1.49) | (0.62 to 2.78) | (0.40 to 2.19) | (0.49 to 4.50) | (1.03 to 4.95) | (–1.25 to 0.33) |
| **RE–respondent acquaintance** | Round 2 2016 | | | | | | | |
| Not acquainted (reference) | – | – | – | – | – | – | – | – |
| Not well acquainted | 1.51* | 1.40 | 1.30 | 1.32 | 1.19 | 1.34 | 1.31 | –0.15 |
| | (1.01 to 2.28) | (0.92 to 2.13) | (0.85 to 1.99) | (0.74 to 2.33) | (0.72 to 1.98) | (0.80 to 2.25) | (0.75 to 2.29) | (–0.54 to 0.24) |
| Well acquainted | 1.55 | 1.07 | 1.79* | 1.35 | 0.60 | 1.48 | 1.89* | –0.48* |
| | (0.98 to 2.45) | (0.70 to 1.64) | (1.10 to 2.89) | (0.72 to 2.52) | (0.35 to 1.01) | (0.80 to 2.75) | (1.02 to 3.49) | (–0.96 to –0.01) |
| Very well acquainted | 1.06 | 0.73 | 1.50 | 1.13 | 0.75 | 1.61 | 1.52 | –0.50 |
| | (0.50 to 2.25) | (0.35 to 1.52) | (0.69 to 3.24) | (0.52 to 2.44) | (0.42 to 1.34) | (0.50 to 5.14) | (0.85 to 2.71) | (–1.08 to 0.07) |
| Interviewed in R1 | 0.62 | 0.63 | 0.77 | 0.81 | 1.37 | 0.72 | 1.00 | 0.46 |
| | (0.36 to 1.05) | (0.31 to 1.27) | (0.45 to 1.34) | (0.45 to 1.46) | (0.70 to 2.70) | (0.37 to 1.39) | (0.56 to 1.77) | (–0.06 to 0.98) |

*p<0.05, **p<0.01.
Models control for age, quadratic age, education, marital status, number of births, wealth quintile and urban/rural residence; SE are clustered by EA.
PMA2020, Performance Monitoring and Accountability 2020 Project; RE, resident enumerator.

**Table 5** Pooled multivariate regression results for the relationship between RE-respondent acquaintance and family planning outcomes, PMA2020 Kongo Central 2015, 2016

| | Using any contraceptive method | | Using a modern contraceptive method | | Using a traditional contraceptive method | | Last birth was unintended | | Experienced the death of a child | | Provided a response to sexual debut | | Reported infertility | | Age at first sex (mean) | |
|---|---|---|---|---|---|---|---|---|---|---|---|---|---|---|---|---|
| | Odds | 95% CI | Odds | 95% CI | Odds | 95% CI | Odds | 95% CI | Odds | 95% CI | Odds | 95% CI | Odds | 95% CI | Coef. | 95% CI |
| RE-respondent acquaintance | | | | | | | | | | | | | | | | |
| Not acquainted (reference) | – | – | – | – | – | – | – | – | – | – | – | – | – | – | – | – |
| Not well acquainted | 1.09 | (0.84 to 1.41) | 1.17 | (0.88 to 1.55) | 0.97 | (0.68 to 1.37) | 1.29 | (0.94 to 1.76) | 0.86 | (0.62 to 1.20) | 0.93 | (0.68 to 1.29) | 1.02 | (0.67 to 1.54) | −0.28 | (−0.56 to 0.01) |
| Well acquainted | 0.36** | (0.17 to 0.76) | 0.43* | (0.19 to 0.99) | 0.58 | (0.21 to 1.59) | 1.43 | (0.61 to 3.32) | 1.77 | (0.75 to 4.22) | 1.68 | (0.66 to 4.29) | 0.78 | (0.26 to 2.34) | 0.11 | (−0.62 to 0.85) |
| Very well acquainted | 1.02 | (0.71 to 1.48) | 0.81 | (0.54 to 1.21) | 1.17 | (0.74 to 1.84) | 1.00 | (0.66 to 1.51) | 0.74 | (0.49 to 1.12) | 1.61* | (1.04 to 2.59) | 1.50 | (0.92 to 2.47) | −0.31 | (−0.68 to 0.05) |
| PMA2020 R2 (2016) | 1.09 | (0.89 to 1.34) | 0.82 | (0.65 to 1.02) | 1.57** | (1.19 to 2.06) | 0.71** | (0.56 to 0.91) | 1.11 | (0.86 to 1.44) | 0.92 | (0.73 to 1.18) | 1.13 | (0.81 to 1.56) | −0.41** | (−0.63 to −0.19) |
| Interviewed in R1 | 0.93 | (0.62 to 1.41) | 0.86 | (0.52 to 1.41) | 1.00 | (0.60 to 1.68) | 1.26 | (0.79 to 2.01) | 1.76* | (1.11 to 2.78) | 0.83 | (0.52 to 1.32) | 0.78 | (0.43 to 1.41) | 0.14 | (−0.30 to 0.59) |
| PMA2020 R2*Well-acquainted interaction | 2.09** | (1.33 to 3.30) | 1.66* | (1.01 to 2.76) | 1.58 | (0.88 to 2.83) | 0.76 | (0.45 to 1.27) | 0.59 | (0.34 to 1.02) | 0.83 | (0.47 to 1.45) | 1.27 | (0.66 to 2.44) | −0.28 | (−0.74 to 0.17) |

*p<0.05, **p<0.01.
Models control for age, quadratic age, education, marital status, number of births, wealth quintile, and urban/rural residence; SE are clustered by EA.
PMA2020, Performance Monitoring and Accountability 2020 Project; RE, resident enumerator.

statistically significant are for the 'well-acquainted' category. Results show evidence for differences in the effect of being well acquainted by PMA2020 survey wave: REs and respondents who were well acquainted in 2016 had greater odds of reporting overall contraceptive use and modern contraceptive use, compared with all others. We also find that women who were interviewed previously in 2015 had greater odds of reporting the death of a child than those not previously interviewed (OR 1.76 (95% CI 1.11 to 2.78)).

## DISCUSSION

We find that the level of acquaintance between interviewer and respondent is significantly associated with a range of self-reported family planning outcomes, including contraceptive use, infertility, age at first sex and last birth unintended. Many of these relationships persist even after controlling for characteristics associated with RE-respondent acquaintance. We also find evidence that the impact of acquaintance on survey responses changes over time for some outcomes, where, for example, individuals who were well acquainted with the RE and interviewed in round 2 (2016) were more likely to report contraceptive use and modern contraceptive use. In contrast, we find very limited evidence that being interviewed by PMA2020 previously has an impact on survey responses in the second round.

Are respondents reporting more or less accurately to REs with whom they are acquainted? Overall, the results suggest that greater acquaintance provides more accurate responses. One might expect that women will under-report sensitive outcomes such as infertility, child mortality and early age at sexual debut. Since greater acquaintance is significantly associated with higher reports of these measures, it is plausible that acquaintance improves respondents' willingness to report a sensitive behaviour. Similarly, research shows that large families are valued and family planning is discouraged in rural areas of DRC,[24] which suggests that women may also under-report contraceptive use. Therefore, higher levels of contraceptive use may represent more accurate responses. However, we do not know the 'true' response for these outcomes and therefore cannot definitively judge if greater familiarity yields better data quality.

The relationship is most common for RE–respondents who are 'well acquainted', instead of 'very well acquainted.' This suggests that while familiarity between RE and respondents has an impact on survey responses, the relationship may be non-linear, where some familiarity is beneficial but too much familiarity may be detrimental to data quality. This may reflect a balance between acquaintance and survey responses, where some familiarity may be beneficial to stimulate an open response, but being too close to the RE may cause the respondent to be fearful of disclosing sensitive information to others in the community. Similarly, there may be ethical implications if the RE and respondent are well-acquainted, and the respondent may be reluctant to disclose sensitive information due to fear of judgement from the RE. This curvilinear relationship is consistent with the literature, which has shown similar patterns in several previous studies of interviewer effects.[2] Overall, there may be benefits to some degree of acquaintance between the interviewer and respondent, but too much familiarity may lead to ethical issues and could be detrimental to data quality.

It is important to note that we cannot claim that the acquaintance between RE and respondent has a causal impact on survey responses. Ideally, one would conduct an experiment, where REs would be randomly assigned to EAs, some who are from the EA and others from outside the EA. In the absence of this, there may be systematic characteristics of REs and EAs that explain this association and are not included in the model. Future research would also benefit from the inclusion of interviewer characteristics (eg, age, education, etc) as potential moderators of this relationship. Of particular interest would be whether the age difference between the RE and respondent impacts survey responses. Another limitation is that acquaintance is asked from the REs perspective, and participants could disagree with the RE about the extent of acquaintance. Other PMA2020 countries employ similar RE recruitment strategies and ask REs to record level of acquaintance with respondents; the generalisability of these findings could be tested by pooling the data for all PMA2020 countries and conducting a cross-country analysis. Despite its limitations, our research provides evidence that acquaintance has a significant impact on survey responses, and is therefore an important source of bias in survey responses.

This research has important implications for survey data collection in settings like DRC. As noted in some studies on data quality, the standard in many large-scale surveys, like DHS, is to use outsider interviewers; PMA2020 is one of the few that uses REs from the local settings. Since we find an association between acquaintance and survey responses, and our results suggest that interviewers from the survey site may yield more accurate responses to family planning outcomes, we recommend that the approach to hiring interviewers be examined and reconsidered by many survey data collection efforts. However, we acknowledge that variation in acquaintance between interviewer and respondent may not exist in some locations, such as urban settings like Kinshasa. To further evaluate the findings here, we recommend that this relationship be tested in other settings, particularly those where contraceptive use is not discouraged. Most importantly, we recommend that the impact of using interviewers from the survey sites be tested using an experimental design, and with outcomes that are validated.

**Author affiliations**
[1]Department of Global Community Health and Behavioral Science, School of Public Health and Tropical Medicine, Tulane University, New Orleans, Louisiana, USA
[2]Faculty of Medicine, Kinshasa School of Public Health, University of Kinshasa, Kinshasa, Congo (the Democratic Republic of the)

[3]Department of Global Health Management and Policy, School of Public Health and Tropical Medicine, Tulane University, New Orleans, Louisiana, USA
[4]Division of Epidemiology and Biostatistics School of Public Health, Kinshasa School of Public Health, University of Kinshasa, Kinshasa, Congo (the Democratic Republic of the)

**Contributors** PAn designed the manuscript, conducted the analysis and wrote the first draft. PAk was responsible for the conception and design, and reviewing all drafts of the manuscript. LPE was responsible for the conception and design, and reviewing all drafts of the manuscript. JH was responsible for the conception and design, and reviewing all drafts of the manuscript. PK managed data collection for PMA2020 in DRC, was responsible for the conception and design, and reviewing all drafts of the manuscript. All authors read and approved the final manuscript.

**Funding** PMA2020 was supported by the Bill & Melinda Gates Foundation, Seattle, WA; under grant #OPP1079004.

**Competing interests** None declared.

**Patient consent** Not required.

**Ethics approval** This study has received approval to collect data from Institutional Review Boards at Johns Hopkins University, Tulane University, and the University of Kinshasa. Participants provided written informed consent to participate in the study.

**Provenance and peer review** Not commissioned; externally peer reviewed.

**Data sharing statement** PMA2020 data used for this analysis is publically available by request (at http://www.pma2020.org/request-access-to-datasets-new).

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
