## [Reviewer comments · BMJ Open]

ARTICLE DETAILS

TITLE (PROVISIONAL)	The relationship between interviewer-respondent familiarity and family planning outcomes in the Democratic Republic of Congo: A Repeat Cross-Sectional Analysis
AUTHORS	Anglewicz, Philip; Akilimali, Pierre; Eitmann, Linnea; Hernandez, Julie; Kayembe, Patrick

VERSION 1 – REVIEW

REVIEWER	Anne Pfitzer Maternal and Child Survival Program/Jhpiego, United States
REVIEW RETURNED	14-Apr-2018

GENERAL COMMENTS	Methods: p.6 top paragraph. This information is probably publically available on PMA2020 website, but you may want to be explicit as to whether randomization of cluster areas and households is repeated for every round, or only the households in each round. Results Table 1. while I didn't check BMJ Open guidelines for tables, I would recommend including Ns with the percentages, to see what categories may have small numbers for the subsequent analyses. Table 2 - p.9 line 8 - Given that fractions of odds ratios are harder to absorb than multiples, authors should consider reversing the reference to avoid the fractions. So that would make urban residence the reference or never married the reference. p9-lines 11-12. The language around who is acquainted with whom seems backwards. From the methods section, it seems the RE answers the question at the top of survey as to whether she is familiar with respondent. It seems the authors assume that acquaintanceship is bidirectional. However, I am not sure that is always the case. For someone who is agnostic, it could be she may not recognize RE even when RE recognizes her. This occurs also in the paragraph directly after this one. Likely this would only affect the "not well acquainted category" but given that these are merged with the well acquainted in the analysis, it could still be more accurate to refer to the RE's acquaintance rather than the respondents. Discussion. It may be useful to look for or review any literature as to the social desirability or stigma associated with contraceptive use in Kongo Central. Stigma around contraception is very culturally specific. In Mozambique, we found little stigma, except in one ethnicity
---

	(unfortunately not yet published). Authors could cite DRC literature to that effect. I was also struck by the lack of consistency in what types of questions had significant evidence of differences types of acquaintances and between rounds. Or that the directionality seems to change. For example, Table 3 implies less willingness to report contraceptive use. In Table 4, only among those not well acquainted and only in the 2nd round, there was higher odds of reporting contraceptive use (but not significant for those better acquainted), then in table 5, the other direction is seen with again lower likelihood of reporting contraceptive use, but only among better acquainted pairs. One would expect more consistency across the various analyses, but it is not really there. The scattershot sprinkling of significance of results in these tables doesn't leave a lot of clarity or confidence that what is seen is really problematic or about what may be more "true". The authors should comment on this. p12 line 16. some words may be missing? between RE "and respondents"... p13 It is not clear what the authors are recommending when they suggest to revisit the RE hiring practices. On the one hand, they say results suggest greater truthfulness, but they acknowledge there is no way to know whether that is the case or not, but the reference to surveys writ large in the recommendation would imply that the PMA approach is recommended. All this ignores the observation in the methods section that data from Kinshasa could not be included in the study because RE were not acquainted with over 98% of respondents. It seems from the Kongo Central data in Table 2, that the same applies to urban areas in that region. So perhaps the recommendation should be limited to rural areas? In any case, any recommendation should leave the reader in no doubt as to what authors are saying. The earlier recommendation of repeating this as a multi-country analysis is also warranted as countries with low stigma for disclosure of contraceptive use may not have the same bias issues. In context where there are social norms of postpartum abstinence, some stigma is associated with postpartum contraceptive use (see Cooper, C et al 2015. GHSP about Liberia). It may be useful to also include in future analyses whether the respondent is within 6-9 months of a birth. There is a typo in line 26 of the abstract where the word infidelity is substituted for the word infertility.
--	---

REVIEWER	Nancy Glass Johns Hopkins University
REVIEW RETURNED	17-Apr-2018

GENERAL COMMENTS	This is an important and interesting study that addresses an important challenge in survey research. I recommend publication with revisions. First, I think it is important to define acquaintance, what is the difference between acquainted (see them and say hi at church for example), well-acquainted (e.g. friend of a friend) and very-well acquainted (e.g. friend)? How was it defined for the RE to make the choice? Did these categories come with examples? In the abstract the conclusion states, "results suggest that greater
---

	familiarity yields better data quality, therefore we recommend that the approach to hiring interviewers be examined and reconsidered in survey data collection efforts." I do not think the researchers can say that the findings from the study demonstrates data quality. Rather, as you noted in discussion, the findings indicate that bias exists in survey research related to acquaintance between RE and women interviewed in the area of contraception/family planning. Also it is interesting that the RE was asked about acquaintance with woman participant, but the woman was not asked about acquaintance with RE. This may have been helpful, could have required RE to answer and then ask the woman the same question about the RE. Look for congruency with responses between RE and participant. The discussion is very basic, there is no mention related potential ethical challenges to RE and woman being well/very well-acquainted? Perhaps some information on the confidentiality/safety and other considerations for interviewing someone that the RE is well/very well-acquainted, especially related to sensitive health and social issues. Perhaps including in the discussion information on the potential risk/ benefits of the approach would add more depth to the discussion and the recommendations for future survey research
--	---

REVIEWER	Guy Stecklov University of British Columbia, Canada
REVIEW RETURNED	05-Jun-2018

GENERAL COMMENTS	This authors are to be commending for making a strong case for the logic of their study. As they argue, a paradigm in data collection has been to ensure that interviewers be “outsiders” with respect to their cultural/community origins relative to the respondents. While some might question whether this approach has been given sufficient attention in Western states, there are more indirect arguments for why this approach may be less problematic. However, assuming that the use of “outsiders” is the appropriate approach in other cultural contexts – whatever they are- is a huge and until very recently ignored assumption. If mistaken, this assumption about the “best” interviewers may have profound effects on the mean and quality of data collected. Given that so much of what is known from LDCs comes from survey data, efforts to investigate this in a systematic manner are seriously needed. This paper is based on an increasingly important and newer set of surveys: PMA2020. These surveys are using state of the art methods but are also allowing some innovative approaches. A key feature in DRC is that they collected information on the degree of familiarity between interviewers and respondents. It is not clear whether this is to be a consistent element in the PMA surveys, and this would be useful to know. The results are impressive and compelling. They point to the importance of considering alternative data collection approaches in LDCs. While they do not have an absolute indicator of data quality, they have strong arguments why the questions they use are strong indicators of data quality. The consistent effect of outsiders on these list of variables all point to clear conclusion: that increased familiarity with interviewers improves data quality.
---

	Two dimension of this study advance prior work. One, that they are working in the DRC. The only experimental study so far has been in the Dominican Republic. The cultural meaning of familiarity and its implications may vary dramatically. Thus, extending the original results to even a second additional setting is of great value. A second advantage here is their use of panel data. This is certainly the first attempt to incorporate panel data in a study of familiarity. That these effects do not change with repeat visits is very interesting and an important contribution. Along with my positive impressions, I have several suggestions. Some may be beyond the scope of this paper but are worth mentioning:  1. Background. I would note in the introduction that scholars sometimes note that the assumption that interviewers are randomly assigned reduces the threat of familiarity (or race/gender etc). It is true that basic regression estimates under certain assumption may not be affected, but just have more noise. One can also think of the bias simply affecting the intercept under particular assumptions. However, this ignores the fact that descriptive statistics are often the most important. Our estimates for example on contraceptive prevalence may be completely off! Related to this – the authors should be clearer in the methodology how they were able to ensure that interviewers did NOT self-select so that they interviewed people they knew or did not know. If this happened, it might affect the validity of the analyses. More discussion on this is required! 2. Terminology: The author begin with familiarity but often shift to a discussion of acquaintance. I find this term more awkward. I'd recommend talking about “degree of familiarity” and “familiarity.” Example on page 4/21 at end of first paragraph: “...may improve with better acquaintance.” This doesn't work in my opinion. It would reader better as “...better familiarity” or “...a greater degree of familiarity.” 3. Methods. Here I'm making a suggestion that is beyond the scope of the paper but I believe would be a useful contribution. Given the panel data, I think the authors might consider a individual level fixed effects. The argument for this is strong, particularly given that repeat visits don't matter. Thus, a test of how changing the degree of familiarity across time for individuals would be a really strong test of their question. 4. Page 12/21: perhaps consider retesting some with a binary indicator of familiarity? Taking the finding a bit farther on a cut-off. The last two comments are mostly suggestions that might be considered in future work. The authors are to be commended on this important contribution.
--	--

VERSION 1 – AUTHOR RESPONSE

Reviewer(s)' Comments to Author:

Reviewer: 1

Methods

p.6 top paragraph. This information is probably publically available on PMA2020 website, but you may want to be explicit as to whether randomization of cluster areas and households is repeated for every round, or only the households in each round.

We appreciate this suggestion, and have clarified on pg 6, which states “The enumeration areas remain the same over time, but new households are selected in each round.”

Results

Table 1. while I didn't check BMJ Open guidelines for tables, I would recommend including Ns with the percentages, to see what categories may have small numbers for the subsequent analyses.

We are concerned that including N's for each cell will make Table 1 difficult to read. But we have included N's for each wave, which can be used to calculate the sample size for each cell.

Table 2 - p.9 line 8 - Given that fractions of odds ratios are harder to absorb than multiples, authors should consider reversing the reference to avoid the fractions. So that would make urban residence the reference or never married the reference.

We agree, and have changed the reference categories in Table 2 accordingly.

p9-lines 11-12. The language around who is acquainted with whom seems backwards. From the methods section, it seems the RE answers the question at the top of survey as to whether she is familiar with respondent. It seems the authors assume that acquaintanceship is bidirectional. However, I am not sure that is always the case. For someone who is agnostic, it could be she may not recognize RE even when RE recognizes her. This occurs also in the paragraph directly after this one. Likely this would only affect the "not well acquainted category" but given that these are merged with the well acquainted in the analysis, it could still be more accurate to refer to the RE's acquaintance rather than the respondents.

It is indeed the case that the RE answers the question about acquaintance at the beginning of the survey. We have reviewed the paper and corrected instances where we mistakenly state the opposite.

We agree that a limitation of the study is that the respondent is not asked about acquaintance with the RE, and this could differ from the RE's perspective. We have stated this limitation on pg 13: “Another limitation is that acquaintance is asked from the REs perspective, and participants could disagree with the RE about the extent of acquaintance.”

Discussion

It may be useful to look for or review any literature as to the social desirability or stigma associated with contraceptive use in Kongo Central. Stigma around contraception is very culturally specific. In Mozambique, we found little stigma, except in one ethnicity (unfortunately not yet published). Authors could cite DRC literature to that effect.

We agree that this is a useful addition. Although there is very limited research on Kongo Central, studies have been conducted elsewhere in DRC that likely apply in this setting. We have added the following sentence to the discussion section: “Research shows that large families are valued and family planning is often discouraged in rural areas of DRC, [24] which suggests that women may also under-report contraceptive use.”

I was also struck by the lack of consistency in what types of questions had significant evidence of differences types of acquaintances and between rounds. Or that the directionality seems to change. For example, Table 3 implies less willingness to report contraceptive use. In Table 4, only among those not well acquainted and only in the 2nd round, there was higher odds of reporting contraceptive use (but not significant for those better acquainted), then in table 5, the other direction is seen with again lower likelihood of reporting contraceptive use, but only among better acquainted pairs. One would expect more consistency across the various analyses, but it is not really there. The scattershot sprinkling of significance of results in these tables doesn't leave a lot of clarity or confidence that what is seen is really problematic or about what may be more "true". The authors should comment on this.

We have edited our results and discussion sections to more accurately explain our results. Some of what appear to be changes in direction in the relationship between acquaintance and various outcomes are in fact due to differences in our analysis between tables.

For example, the reason why Table 3 shows less contraceptive use with greater acquaintance is because contraceptive use is associated with many characteristics that are also associated with RE-responder acquaintance (e.g., age, education, etc...); after controlling for these characteristics in multivariate results shown in Table 4, the relationship between acquaintance and contraceptive use is clearer. To clarify this, we state "After controlling for characteristics associated both with RE-responder acquaintance and contraceptive use, such as age and urban residence, our results differ from those shown in Table 2."

In Table 4, we also find that odds of contraceptive use are higher for modern methods among the well-acquainted in round 2.

In Table 5, our focus is on the interaction between PMA2020 round 2 and well-acquainted ("PMA2020 R2*Well-acquainted interaction"), which shows, for example, that REs-respondents who are well acquainted in round 2 have 2.09 greater odds of using contraception. Because of the interaction term, the main well-acquainted ("well acquainted") is *not* interpreted as the relationship between well-acquainted and contraceptive use. Instead, it is the relationship between well-acquainted and contraceptive use when Round 2 is zero, compared to when round 2 is one. In other words, the main effect for well acquainted in Table 5 shows that the well acquainted have smaller odds of using contraception in round 1 compared to round 2. Because we test the relationship between well acquainted and contraceptive use for each round in previous analysis (results shown in Table 4), the main effects are not of interest to our results here. To clarify this point, we have changed the font for results that are not of interest in Table 5, removing them from bold font to plain font.

p12 line 16. some words may be missing? between RE "and respondents"...

We have corrected this error.

p13 It is not clear what the authors are recommending when they suggest to revisit the RE hiring practices. On the one hand, they say results suggest greater truthfulness, but they acknowledge there is no way to know whether that is the case or not, but the reference to surveys writ large in the recommendation would imply that the PMA approach is recommended. All this ignores the observation in the methods section that data from Kinshasa could not be included in the study because RE were not acquainted with over 98% of respondents. It seems from the Kongo Central data in Table 2, that the same applies to urban areas in that region. So perhaps the recommendation should be limited to rural areas? In any case, any recommendation should leave the reader in no doubt as to what authors are saying.

We agree with these points, and have revised the final paragraph of the discussion section to clarify our overall recommendations. In short, we acknowledge that variation in acquaintance between interviewer and respondent may not exist in some settings, like Kinshasa. Most importantly, we also recommend that the relationship between interviewer-respondent acquaintance and various outcomes be tested using experimental design, and with outcomes that can be validated.

The earlier recommendation of repeating this as a multi-country analysis is also warranted as countries with low stigma for disclosure of contraceptive use may not have the same bias issues. In context where there are social norms of postpartum abstinence, some stigma is associated with postpartum contraceptive use (see Cooper, C et al 2015. GHSP about Liberia). It may be useful to also include in future analyses whether the respondent is within 6-9 months of a birth.

We agree, and have added the following statement to the discussion section: "To further evaluate the findings here, we recommend that this relationship be tested in other settings, particularly those where contraceptive use is not discouraged."

There is a typo in line 26 of the abstract where the word infidelity is substituted for the word infertility.

We have corrected this error.

Reviewer: 2

This is an important and interesting study that addresses an important challenge in survey research. I recommend publication with revisions.

First, I think it is important to define acquaintance, what is the difference between acquainted (see them and say hi at church for example), well-acquainted (e.g. friend of a friend) and very-well acquainted (e.g. friend)? How was it defined for the RE to make the choice? Did these categories come with examples?

We appreciate this point. "Acquaintance" was clearly defined in the training manual for REs under PMA2020. The full definition from the PMA202 RE training manual is below. To clarify this, we have added the following statement to the "Measures" section of the paper (pg 6):

"Very well acquainted" was defined as the RE knowing the respondent's first name and would greet her if they met at the market, church, or mosque; "well acquainted" was that the RE knows the respondent by sight and may know a family member as well; "not well acquainted" is that the RE has seen the woman at community or church functions but does not know her name or does not recognize her but knows someone else in the household; "not acquainted" as the RE has never seen the respondent or anyone in her family before."

Definition of "acquaintance" from PMA2020 RE training manual:

FQ G: ACQUAINTANCE

It is possible that you will know one or more of the respondents whom you will be interviewing. Before you go to the informed consent, enter how well you know the respondent.

- Select "VERY WELL ACQUAINTED" if you know the respondent's first name and would greet her if you met at the market, church, or mosque.
- Select "WELL ACQUAINTED" if you know the respondent by sight and may know a family member well.

- Select "NOT WELL ACQUAINTED" if you have seen the woman at community or church functions but do not know her name or if you do not recognize her but do know someone else in the household.
- Select "NOT ACQUAINTED" if you have never seen the respondent or anyone in her family before.

In the abstract the conclusion states "results suggest that greater familiarity yields better data quality, therefore we recommend that the approach to hiring interviewers be examined and reconsidered in survey data collection efforts." I do not think the researchers can say that the findings from the study demonstrates data quality. Rather, as you noted in discussion, the findings indicate that bias exists in survey research related to acquaintance between RE and women interviewed in the area of contraception/family planning.

We agree, and have rephrased as "Our results show that interviewer-respondent acquaintance is associated with a range of family planning outcomes; therefore we recommend that the approach to hiring interviewers be examined and reconsidered in survey data collection efforts."

Also it is interesting that the RE was asked about acquaintance with woman participant, but the woman was not asked about acquaintance with RE. This may have been helpful, could have required RE to answer and then ask the woman the same question about the RE. Look for congruency with responses between RE and participant.

We agree that a limitation of the study is that the respondent is not asked about acquaintance with the RE, and this could differ from the RE's perspective. We have stated this limitation on pg 13: "Another limitation is that acquaintance is asked from the REs perspective, and participants could disagree with the RE about the extent of acquaintance."

The discussion is very basic, there is no mention related potential ethical challenges to RE and woman being well/very well-acquainted? Perhaps some information on the confidentiality/safety and other considerations for interviewing someone that the RE is well/very well-acquainted, especially related to sensitive health and social issues. Perhaps including in the discussion information on the potential risk/ benefits of the approach would add more depth to the discussion and the recommendations for future survey research.

We agree, and have expanded our discussion section to address the above point on pgs 12-13: "Similarly, there may be ethical implications if the RE and respondent are well-acquainted, and the respondent may be reluctant to disclose sensitive information due to fear of judgement from the RE. This curvilinear relationship is consistent with the literature, which has shown similar patterns in several previous studies of interviewer effects.[2] Overall, there may be benefits to some degree of acquaintance between the interviewer and respondent, but too much familiarity may lead to ethical issues and could be detrimental to data quality."

Reviewer: 3

The authors are to be commending for making a strong case for the logic of their study. As they argue, a paradigm in data collection has been to ensure that interviewers be "outsiders" with respect to their cultural/community origins relative to the respondents. While some might question whether this approach has been given sufficient attention in Western states, there are more indirect arguments for why this approach may be less problematic. However, assuming that the use of "outsiders" is the appropriate approach in other cultural contexts – whatever they are- is a huge and until very recently ignored assumption. If mistaken, this assumption about the "best" interviewers may have profound effects on the mean and quality of data collected. Given that so much of what is known from LDCs comes from survey data, efforts to investigate this in a systematic manner are seriously needed.

This paper is based on an increasingly important and newer set of surveys: PMA2020. These surveys are using state of the art methods but are also allowing some innovative approaches. A key feature in DRC is that they collected information on the degree of familiarity between interviewers and respondents. It is not clear whether this is to be a consistent element in the PMA surveys, and this would be useful to know.

Acquaintance between RE and respondent was included in surveys for all PMA2020 countries. We have added this point to the “measures” section of the paper (pg 6).

The results are impressive and compelling. They point to the importance of considering alternative data collection approaches in LDCs. While they do not have an absolute indicator of data quality, they have strong arguments why the questions they use are strong indicators of data quality. The consistent effect of outsiders on these list of variables all point to clear conclusion: that increased familiarity with interviewers improves data quality.

Two dimension of this study advance prior work. One, that they are working in the DRC. The only experimental study so far has been in the Dominican Republic. The cultural meaning of familiarity and its implications may vary dramatically. Thus, extending the original results to even a second additional setting is of great value.

A second advantage here is their use of panel data. This is certainly the first attempt to incorporate panel data in a study of familiarity. That these effects do not change with repeat visits is very interesting and an important contribution.

We appreciate the positive comments about this research.

Along with my positive impressions, I have several suggestions. Some may be beyond the scope of this paper but are worth mentioning:

1. Background. I would note in the introduction that scholars sometimes note that the assumption that interviewers are randomly assigned reduces the threat of familiarity (or race/gender etc). It is true that basic regression estimates under certain assumption may not be affected, but just have more noise. One can also think of the bias simply affecting the intercept under particular assumptions. However, this ignores the fact that descriptive statistics are often the most important. Our estimates for example on contraceptive prevalence may be completely off!

We agree, and now state on pg 3: “This potential source of bias is often ignored entirely. Other times, interviewer characteristics are seen as part of the error term in multivariate regression and assumed to not impact estimates, despite the fact that these characteristics may in fact be associated with outcomes of interest, and descriptive statistics (like contraceptive use) are often of primary interest.”

Related to this – the authors should be clearer in the methodology how they were able to ensure that interviewers did NOT self-select so that they interviewed people they knew or did not know. If this happened, it might affect the validity of the analyses. More discussion on this is required!

We agree with this point, and now state on pg 6: “Participating households were selected by the central survey management team, not the RE, which ensures that REs did not systematically select households with friends or acquaintances.”

2. Terminology: The authors begin with familiarity but often shift to a discussion of acquaintance. I find this term more awkward. I’d recommend talking about “degree of familiarity” and “familiarity.” Example

on page 4/21 at end of first paragraph: "...may improve with better acquaintance." This doesn't work in my opinion. It would reader better as "...better familiarity" or "...a greater degree of familiarity."

We agree that "familiarity" is a clearer way of describing the relationship of interest in this research. However, the phrasing of the question in the PMA2020 surveys is "acquaintance", so we prefer to refer to the relationship as such to accurately portray the measure. That said, we also agree that the above example was awkwardly phrased, and we have revised as "a greater degree of familiarity".

3. Methods. Here I'm making a suggestion that is beyond the scope of the paper but I believe would be a useful contribution. Given the panel data, I think the authors might consider a individual level fixed effects. The argument for this is strong, particularly given that repeat visits don't matter. Thus, a test of how changing the degree of familiarity across time for individuals would be a really strong test of their question.

We agree, and appreciate this point. Since only about 10% of respondents were interviewed in both waves, the sample size for these two waves may not be sufficient for individual-level fixed effects analysis (which would drop anyone who doesn't change in outcomes over time). But this approach may be possible after another wave of data collection in Kongo Central, which would presumably include another 10% or more of individuals who were interviewed in a previous wave.

4. Page 12/21: perhaps consider retesting some with a binary indicator of familiarity? Taking the finding a bit farther on a cut-off.

We tested this relationship with a binary measure of "not acquainted" compared to "very well acquainted", "well acquainted", "not well acquainted". The results were not substantively different from those found here.

However, a binary measure of acquaintance does not permit analysis of the curvilinear relationship between acquaintance and accurate responses, where some acquaintance may be beneficial but too much familiarity may be detrimental, which we find particularly interesting, so we see value in keeping this as a four category measure.

The last two comments are mostly suggestions that might be considered in future work. The authors are to be commended on this important contribution.

We thank the reviewer for their useful comments on our research.

VERSION 2 – REVIEW

REVIEWER	Anne Pfitzer Jhpiego, United States
REVIEW RETURNED	23-Aug-2018

GENERAL COMMENTS	The authors have thoughtfully addressed all the reviewer comments including my own and made the right edits to their manuscript. I have no further comments.
--

REVIEWER	Nancy Glass Johns Hopkins University, USA
REVIEW RETURNED	04-Sep-2018

GENERAL COMMENTS	The authors provided adequate responses and clarification to reviews and I have no further concerns prior to publication. I think the study is an important contribution to the literature.
REVIEWER	Guy Stecklov University of British Columbia, Canada
REVIEW RETURNED	24-Aug-2018
GENERAL COMMENTS	The revisions of this paper make it ready for acceptance.

VERSION 2 – AUTHOR RESPONSE

Thank you for the opportunity to revise my manuscript for consideration by BMJ Open. In response to the most recent comments, I have made the following revisions:

1. I have added the study design to the title of the manuscript.
2. I have added a "Design" sub-section to the abstract.
3. I have revised the "Patient and Public Involvement" statement.

Please let me know if any further edits or revisions are needed.